# Gain Function-Based Visual Tracking Control for Inertial Stabilized Platform with Output Constraints and Disturbances

**Xiangyang Liu** [1,*] **, Jun Yang** [2,*] **and Pengyu Qiao** [1]

1   School of Automation, Southeast University, Nanjing 210096, China; qpy@seu.edu.cn
2   Department of Aeronautical and Automotive Engineering, Loughborough University,
    Loughborough LE11 3TU, UK
*   Correspondence: xy_liu@seu.edu.cn (X.L.); j.yang3@lboro.ac.uk (J.Y.)

**Abstract:** In this paper, a composite control strategy is proposed to deal with output constraints and disturbances of the visual tracking system for an inertial stabilized platform, which combines active disturbance compensation and the variable gain function technique. Firstly, the model of system considering multi-source disturbances is established, where the controlled output is the constrained position of the target in the image plane. Secondly, in order to avoid the tracked target being lost in the field of view of the camera, a control method based on the variable gain function technique is designed to ensure that the controlled output remains within the feasible range. Moreover, the active disturbance estimation and compensation method is introduced to improve the anti-disturbance ability of the system under the situation of small output error, obtaining satisfactory tracking performance. The stability analysis and the proof of constrained output are given following the controller design. Finally, results of simulation and experiments are shown to illustrate the promised advantages of the proposed composite control approach.

**Keywords:** inertial stabilized platform; visual tracking control; output constraint; disturbance observer; variable gain function





## 1. Introduction

Vision-based tracking control is widely used in many unmanned autonomous systems (UASs), such as mobile robots and unmanned aerial vehicles, due to the strong environmental perception [1–4]. The task of a visual tracking system is to ensure that the optical equipment always aims at the target; in other words, the tracked target remains in the center of the image [5]. For outdoor UAS, the optical equipment installed on a mobile transporter is greatly affected by the vibration of the carrier, which reduces the imaging quality and aiming accuracy of the optical instrument. In order to overcome this problem, the inertial stabilized platform (ISP) is introduced to isolate the disturbance from the carrier and ensure the inertial stability of the optical equipment [6–8], which uses a gyroscope to measure the angular velocity of the optical instrument with respect to the Earth. Therefore, the visual tracking system based on an ISP has been investigated by many scholars [9–12]. In visual tracking control, avoiding the loss of the target from the field of view of the camera is a key technology to realize the tracking task.

The research on the target loss problem of a visual tracking system can be divided into passive and active strategies [13–15]. Passive methods focus on the trajectory prediction of the target [16,17]; however, due to the inaccurate kinematic model and target maneuver, these methods are unable to achieve effective position prediction in the case of long-term losing, and lead to the failure of the target tracking task. Meanwhile, active approaches aim to ensure that the position of the target is constrained within the feasible region through the designed controllers [18–20], which transform the target loss problem into output constraint control. It is noted that output constraints are inevitable in practical systems due to limited hardware conditions. To deal with this issue, an approach based on the barrier Lyapunov function

(BLF) is adopted [21]. In [22], a tan-type BLF is introduced into the energy function to solve the position constraints problem of the manipulator. In [23], a log-type BLF is employed for the manipulator with full-state constraints. In [24], an ln-type BLF is adopted for a robot with output constraints. However, the penalty terms added in the above control methods reduce the dynamic performance of systems. The model predictive control (MPC) is another method to solve the constraint problem in a system [25,26]. The core idea of MPC is to obtain a sub-optimal solution of the control input that meets the cost function with system constraints [27]. However, the complex calculation results in a long update cycle of the digital controller, which limits the fast tracking performance of the visual tracking system [28,29].

A practical approach with a simple structure is the gain function-based controller to deal with the system constraints [30,31]. Specifically, by introducing a time-varying gain function, the controller has a large feedback gain when the output error is large, and generates an appropriate gain when the error is small to suppress the influence of measurement noise. In [32], a phase-based variable gain feedback control is utilized to improve the settling performance of precision motion systems. In [33], a variable gain-prescribed performance control law is proposed for the dynamic positioning of ships with positioning error constraints and input saturation. However, for the disturbed system, this method cannot guarantee satisfactory anti-disturbance performance [34]. There are two main reasons for this issue. First, the feedback gain is too small to compromise the noise amplification and the disturbance rejection when the output error is small, so that the control effect is not sufficient to counteract the disturbance in the system. Second, this control strategy produces a corresponding control effect only after the disturbance, resulting in the delayed suppression of the disturbance.

Although the image-based visual tracking scheme is robust to the system uncertainties, applying this method to an ISP system still suffers from multiple disturbances [35]. The tracking accuracy of the ISP visual system is substantially affected by various kinds of external disturbances and internal uncertainties, such as the uncertain motion of the target, the vibration of the carrier, the uncontrollable angular velocity of the camera, and the unknown time-varying feature depth of the target [36]. Furthermore, the visual tracking system should possess more robust performance against the tracking error of desired inertial angular velocity from the inner stabilized loop due to the friction torque and the mass imbalance [37]. Sometimes, the visual tracking system is affected by large disturbances, such as the rapid movement of the target and the large-scale vibration of the carrier, which may lead to the target position being close to the boundary.

The disturbance rejection method is widely used in engineering, where the lumped disturbance is estimated for feed-forward compensation. Many approaches have been proposed to estimate disturbances, including the unknown input observer (UIO) [38], the disturbance observer (DOB) [39], the equivalent input disturbance approach (EID) [40], and the extended state observer (ESO) [41]. The ESO approach is considered to be an effective method in industry. However, the commonly linear ESO can only completely suppress a constant disturbance. Furthermore, the generalized proportional integral observer (GPIO) approach can realize the accurate compensation of the disturbance in polynomial form [42,43]. In the ISP visual tracking system, the polynomial-form disturbance is a major component, so the GPIO method is designed in this system. The GPIO method is similar to a patch, improving the anti-disturbance ability of the variable gain controller when the output error is small. However, the disturbance compensation method cannot ensure that the system output does not cross the boundary. This is because the disturbance observation is a dynamic process. It takes a certain time from the occurrence of the disturbance to accurate estimation; that is, it cannot produce a sufficient control effect in time. In conclusion, a controller design based upon the combination of an effective disturbance rejection method and gain function technology is of great significance to this system with output constraints and disturbances.

In this paper, a composite control strategy combining the disturbance observer and variable gain function is proposed to realize the output constraint control for the ISP

visual tracking system with disturbances. Firstly, the visual tracking system of an ISP considering multi-source disturbances is modeled, where the controlled output is the constrained position of the target in the image plane. Then, the GPI observer is designed to compensate for the lumped disturbance in the system, and the variable gain function is introduced in the controller to ensure that the system output is always in the given range. Stability analysis is given to theoretically analyze the proposed control method. Finally, the results of the simulation and experiments are shown to illustrate the advantages of the gain function-based approach and the proposed composite control method. The proposed strategy has the following advantages:

1. The gain function-based controller is designed to ensure that the system output is constrained to a feasible region to prevent the target loss;
2. An active disturbance rejection method is introduced to enhance the high-precision tracking performance of the system when the output error is small.

The remaining parts of the paper are organized as follows. The model of the ISP visual tracking system with output constraints and disturbances is established in Section 2. Section 3 shows the design of the proposed composite control strategy and the stability analysis. Then, Section 4 gives the simulations and experimental results. Finally, the research conclusions are summarized in Section 5.

## 2. Modeling of ISP Visual Tracking System

Modeling of the system and problem description are to be given in this section. The relationship between the controlled output and the control input of this system is characterized by the kinematics theory. According to the hardware configuration and engineering requirements, the control problem description is introduced.

### 2.1. Visual Tracking System Kinematics

In order to design a tracking controller for this system, the model between the system input (the inertial angular velocity of camera) and the system output (the position of target in the image) should be established first.

To analyze the kinematic relationship between the system output and control input, the camera frame $\{C\}$ and the Earth frame $\{E\}$ are introduced to construct the reference coordinate system of the visual tracking system, as shown in Figure 1. The coordinates of the target with respect to the camera frame $\{C\}$ and Earth frame $\{E\}$ are defined as $P_c = [x_c, y_c, z_c]^T$ and $P_e = [x_e, y_e, z_e]^T$, respectively; the position of the camera frame $\{C\}$ with respect to the Earth frame $\{E\}$ is denoted by $P_c^e = [x_c^e, y_c^e, z_c^e]^T$, and the rotation matrix from the camera frame $\{C\}$ to the Earth frame $\{E\}$ is defined as $R_c^e$. Moreover, the pixel coordinate of the target is given by $X = [x_1, x_2]^T$ in the image plane, and the origin of the pixel coordinate system is specified in the geometric center of the image. According to the imaging principle and perspective projection theory, the target position $X$ in the image is calculated as

$$X = \frac{1}{z_c} \begin{bmatrix} k\lambda & 0 & 0 \\ 0 & k\lambda & 0 \end{bmatrix} P_c, \tag{1}$$

where $k$ is the scale factor between the pixel and the unit meter, and $\lambda$ is the focal length of the camera. Furthermore, the time derivative of the target position $X$ is calculated as

$$\dot{X} = L(z_c)\dot{P}_c, \tag{2}$$

in which

$$L(z_c) = \frac{1}{z_c} \begin{bmatrix} k\lambda & 0 & -x_1 \\ 0 & k\lambda & -x_2 \end{bmatrix}.$$

The above equation establishes the relationship between the line velocities of the target in the pixel coordinate system and the camera coordinate system, and then the relationship between the target position in the camera coordinate system and the inertial angular velocity of the camera is to be built. $W_c = [w_x, w_y, w_z]^T$ and $V_c = [v_x, v_y, v_z]^T$ are

defined as the inertial angular velocity and the line velocity of the camera with respect to the camera frame $\{C\}$, and $W_e$ denotes the inertial angular velocity of the camera with respect to the Earth frame $\{E\}$. Then, the following kinematics can be presented as

$$P_c = (R_c^e)^{\mathrm{T}}(P_e - P_c^e),$$
$$\dot{P}_c^e = R_c^e V_c, \quad \dot{R}_c^e = S(W_e)R_c^e, \quad W_e = R_c^e W_c, \tag{3}$$

where $S$ is the operation symbol of the skew symmetric matrix. The derivative of target position $P_c$ expressed in camera frame $\{C\}$ is given by

$$\dot{P}_c = (\dot{R}_c^e)^{\mathrm{T}}(P_e - P_c^e) + (R_c^e)^{\mathrm{T}}(\dot{P}_e - \dot{P}_c^e). \tag{4}$$

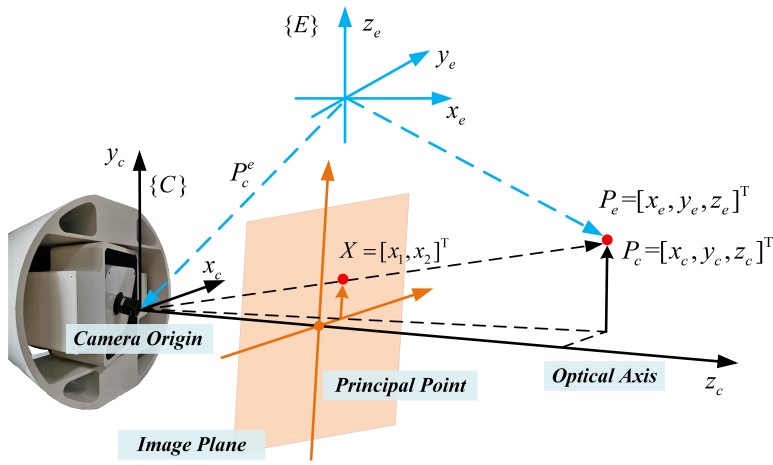

**Figure 1.** Reference coordinate system of the visual tracking system for an ISP.

Combining Equation (3), it has the dynamic between the target position $P_c$ and the inertial angular velocity of the camera $W_c$ as

$$\begin{aligned}
\dot{P}_c &= (R_c^e)^{\mathrm{T}}[S^{\mathrm{T}}(W_e)(P_w - P_c^e)] + (R_c^e)^{\mathrm{T}}\dot{P}_e - V_c \\
&= -(R_c^e)^{\mathrm{T}}[W_e \times (P_e - P_c^e)] + (R_c^e)^{\mathrm{T}}\dot{P}_e - V_c \\
&= -W_c \times P_c + (R_c^e)^{\mathrm{T}}\dot{P}_e - V_c.
\end{aligned} \tag{5}$$

In order to obtain the relationship between the target position in the image and the inertial angular velocity of the camera by substituting (5) into (2), one obtains

$$\dot{X} = L_w W_c + L_v(z_c)[(R_c^e)^{\mathrm{T}}\dot{P}_e - V_c], \tag{6}$$

where

$$L_w = \frac{1}{k\lambda}\begin{bmatrix} x_1 x_2 & -(k\lambda)^2 - x_1^2 & k\lambda x_2 \\ (k\lambda)^2 + x_2^2 & -x_1 x_2 & -k\lambda x_1 \end{bmatrix}.$$

In fact, for a two-axis ISP used in this system, only the angular velocities in two directions can be measured and adjusted. Therefore, the angular velocities $w_x$ and $w_y$ of $W_c$ are available for the control inputs in the presented hardware configuration, so that we rearrange system (6) as

$$\dot{X}(t) = L_u U_x + D(t), \tag{7}$$

where $U_x = [w_x, w_y]^{\mathrm{T}}$ is taken as the control input, and $L_u$ is expressed by

$$L_u = \frac{1}{k\lambda}\begin{bmatrix} x_1 x_2 & -(k\lambda)^2 - x_1^2 \\ (k\lambda)^2 + x_2^2 & -x_1 x_2 \end{bmatrix},$$

and $D(t)$ is regarded as the lumped disturbance of the system arising from the unknown visual depth of target $z_c$, the line velocity of the camera $V_c$, the line velocity of target $\dot{P}_e$, and

the inertial angular velocity $w_z$. According to previous research, the lumped disturbance in this system mainly comes from the target motion and the outdoor carrier. Although these two kinds of disturbance are unknown, it can be found that their main components are polynomial, which is of profound significance for the later disturbance estimation and compensation.

### 2.2. Control Objective

In fact, the value of $k\lambda$ is so large that it becomes the main part of the matrix $L_u$. To simplify the design and implementation of the visual tracking controller, the nominal matrix $L_u^*$, which only retains constant parts of matrix $L_u$, is adopted. Thus, the state-space model of the ISP visual tracking system (7) is rewritten as

$$\begin{cases} \dot{X}(t) = L_u^* U_x + D(t), \\ Y(t) = X(t), \end{cases} \tag{8}$$

where $Y(t)$ is the measurement of the system output.

The control objective of this system is to make the target position $X(t)$ in the image plane converge to a bounded region in spite of the presence of lumped disturbance $D(t)$, and be always within the given range. To simplify the controller design, a virtual control input $U(t) = L_u^* U_x$ is defined, and then system (8) is rewritten as

$$\begin{cases} \dot{X}(t) = U(t) + D(t), \\ Y(t) = X(t). \end{cases} \tag{9}$$

In this system, the image size is $640 \times 480$ pixels; therefore, the outputs of the system are constrained with $x_1 \in [0, 640]$ and $x_2 \in [0, 480]$. When the target is lost in the field of view of the camera, this will be another control problem for the trajectory prediction. The problem discussed in this paper is to strictly limit the controlled output of the system to the above ranges by designing a controller based on the gain function technique.

Furthermore, in order to simplify the later derivation and controller design, the center point of the image is specified as the geometric center. That is, the constrained output ranges of the system are $x_1 \in [-320, 320]$ and $x_2 \in [-240, 240]$.

## 3. Design of Controller with Output Constraints

In this section, the disturbance rejection and output constraint are considered. Firstly, the lumped disturbance estimation based on GPIO is designed to improve the anti-disturbance ability of the system with a small output error. Then, the controller based on the gain function is developed to ensure that the target is always in the field of view of the camera. Finally, stability analysis and proof of output constraint are given to verify the proposed method.

### 3.1. Disturbance Observer Design

In order to improve the anti-disturbance ability and the tracking accuracy of the ISP visual tracking system, a disturbance observer is to be designed. The disturbance in this visual tracking system mainly comes from the movement of the target and the motion of the carrier. These two kinds of disturbance accord with the polynomial form. Therefore, a GPIO is to be designed to improve the anti-disturbance ability of the system. For system (9), we define the derivative of lumped disturbance as $H(t) = [h_1, h_2]^\mathsf{T}$, and then the linear GPIO is designed as

$$\begin{cases} \dot{\hat{X}}(t) = U(t) + \hat{D}(t) - L_1(\hat{X}(t) - Y(t)), \\ \dot{\hat{D}}(t) = \hat{H}(t) - L_2(\hat{X}(t) - Y(t)), \\ \dot{\hat{H}}(t) = -L_3(\hat{X}(t) - Y(t)), \end{cases} \tag{10}$$

where $\hat{X}(t) = [\hat{x}_1, \hat{x}_2]^T$, $\hat{D}(t) = [\hat{d}_1, \hat{d}_2]^T$, and $\hat{H}(t) = [\hat{h}_1, \hat{h}_2]^T$ are estimates of the system state $X(t)$, the lumped disturbance $D(t)$, and its derivative $H(t)$, respectively, and $L_1$, $L_2$ as well as $L_3$ represent designed observer gain matrices with appropriate dimensions.

**Assumption 1.** *The second derivative of lumped disturbance $\dot{H}(t)$ of system (9) is bounded.*

It can be found that the lumped disturbance of system (9) consists of the unknown feature depth $z_c$, the line velocity of target $\dot{P}_e$, the line velocity of camera $V_c$, and the angular velocity of camera $w_z$. In fact, these environmental factors $z_c$, $\dot{P}_e$, $V_c$, and $w_z$ are unknown and time-varying, but also bounded due to the limited energy in the real physical world. Therefore, the second derivative of the lumped disturbance $\dot{H}(t)$ is also bounded in practice.

**Theorem 1.** *Under Assumption 1, the disturbance estimate error of observer (10) asymptotically converges to a bounded region, by selecting the appropriate observer gain matrices.*

**Proof of Theorem 1.** Define the state estimate error as $e_X = \hat{X} - X$, the disturbance estimate error as $e_D = \hat{D} - D$, and the estimate error of the disturbance derivative as $e_H = \hat{H} - H$, respectively. Then, combining system (9) and observer (10), one obtains the estimate error dynamics as

$$\begin{cases} \dot{e}_X(t) = e_D(t) - L_1 e_X(t), \\ \dot{e}_D(t) = e_H(t) - L_2 e_X(t), \\ \dot{e}_H(t) = -L_3 e_X(t) - \dot{H}(t). \end{cases} \tag{11}$$

Furthermore, dynamics (11) can be described as

$$\begin{bmatrix} \dot{e}_X(t) \\ \dot{e}_D(t) \\ \dot{e}_H(t) \end{bmatrix} = \underbrace{\begin{bmatrix} -L_1 & I & O \\ -L_2 & O & I \\ -L_3 & O & O \end{bmatrix}}_{\Lambda} \begin{bmatrix} e_X(t) \\ e_D(t) \\ e_H(t) \end{bmatrix} - \begin{bmatrix} O \\ O \\ I \end{bmatrix} \dot{H}(t). \tag{12}$$

Since the second derivative of the lumped disturbance is bounded, it can be found that the observer estimate error asymptotically converges to a bounded region by selecting appropriate observer gain matrices to ensure that $\Lambda$ is Hurwitz. □

The selection of observer gain is specifically explained here to facilitate the engineering implementation for readers. A large observer gain reduces the disturbance estimate error, but also results in noise amplification. Therefore, the selection is a compromise based on the actual effect, in noise amplification and observation error.

### 3.2. Gain Function-Based Controller Design with Disturbance Compensation

To begin with, the following two constraint functions are constructed as gain functions for the controller design, expressed as

$$\omega_1(x_1) = \frac{320^2}{(320 - x_1)^2} + \frac{320^2}{(-320 - x_1)^2}, \tag{13}$$

$$\omega_2(x_2) = \frac{240^2}{(240 - x_2)^2} + \frac{240^2}{(-240 - x_2)^2}. \tag{14}$$

**Theorem 2.** *For system (9), combining disturbance observer (10), if the initial position of the target satisfies $x_1 \in [-320, 320]$ and $x_2 \in [-240, 240]$, a robust control law is designed as*

$$U(t) = -K\Omega X(t) - \hat{D}(t), \tag{15}$$

*where*

$$K = \begin{bmatrix} k_1 & 0 \\ 0 & k_2 \end{bmatrix}, \Omega = \begin{bmatrix} \omega_1(x_1) & 0 \\ 0 & \omega_2(x_2) \end{bmatrix},$$

*in which $k_1, k_2 > 0$ are positive constants, so that the closed-loop system is stable and the system output is always in the given region.*

The diagram of the proposed control scheme is shown in Figure 2.

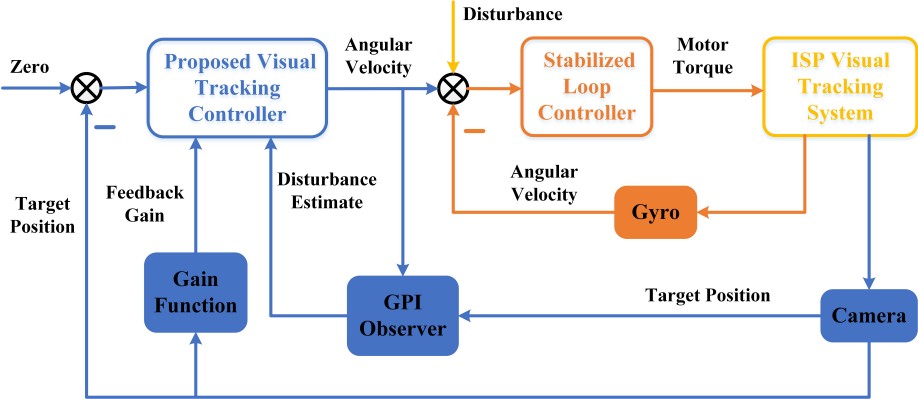

**Figure 2.** The control block diagram of the proposed method based on GPIO and gain function.

*3.3. Stability Analysis and Proof of Constrained Output*

**Proof of Theorem 2.** By substituting disturbance observer (10) and control input (15) into system (9), the closed-loop dynamic of the system is given by

$$\dot{X}(t) = -K\Omega X(t) - \hat{D}(t) + D(t). \tag{16}$$

The analysis will be divided into two parts. First, the system output is proven to converge to a bounded region. Second, the system output never exceeds the given boundary. □

3.3.1. Stability Analysis

The system without output constraints will be proven to be bounded stable for the initial state $X(0)$ firstly. System (16) is rewritten as

$$\dot{X}(t) = -K\Omega X(t) - e_D, \tag{17}$$

where $e_D = [e_{D1}, e_{D2}]^\mathrm{T}$ is the bounded estimate error of the lumped disturbance.

Without losing generality, the stability of state $x_1$ is analyzed separately to simplify the derivation process. According to system (17), choose a energy function as

$$V(t) = \frac{1}{2}x_1^2. \tag{18}$$

It is obvious that the function $V(t)$ is positive definite.

Along system (17), the time derivative of function $V(t)$ is calculated as

$$\dot{V}(t) = \dot{x}_1 x_1 = -k_1 \omega_1 x_1^2 - e_{D1} x_1. \tag{19}$$

The stability of the output $x_1$ is to be analyzed in two steps. First, for initial position $x_1(0)$ (outside the region), the output reaches the boundary in a finite time. Second, the output is always maintained in a bounded region when it arrives.

According to Equation (19), one has $\dot{V}(t) < 0$ when the output $x_1(t)$ is out of range $[-e_{D1}^*/k_1\omega_{1min}, e_{D1}^*/k_1\omega_{1min}]$, where $e_{D1}^*$ represents the boundary of lumped disturbance

$e_{D1}$ and $\omega_{1min}$ represents the minimum value of gain function $\omega_1$, respectively. Therefore, the output $x_1(t)$ converges to the region $[-e_{D1}^*/k_1\omega_{1min}, e_{D1}^*/k_1\omega_{1min}]$ from the initial position $x_1(0)$ in a finite time.

Since it has $\dot{V}(t) \leq 0$ when the output $x_1$ is on the boundary, the output will never exceed the region when it arrives. For the situation of the initial position within the range, the output $x_1$ obviously has the same stability result. Because the output $x_2$ and the output $x_1$ have the same structure and characteristics, the system output $X(t)$ is bounded stable with designed controller (15) and disturbance observer (10).

### 3.3.2. Proof of Output Constraints

In this part, it is shown that the positions $x_1$ and $x_2$ always maintain the preset constrained regions under control law (15) and disturbance observer (11), with the initial condition satisfying $x_1(0) \in [-320, 320]$ and $x_2(0) \in [-240, 240]$. Here, it is still only proven in detail that the state $x_1(t)$ is always within the limited range.

Without loss of generality, assume that there exists a time such that the position $x_1$ reaches the preset maximum 320. According to Equation (19), one has

$$\lim_{x_1 \to 320} \dot{V}(t) = \lim_{x_1 \to 320} -k_1\omega_1(x_1)x_1^2 - e_{D1}x_1 = -\infty. \tag{20}$$

Since it has $\omega_1 \to +\infty$ when $x_1 \to 320$ according to definition (13), the time derivative of the energy function contains a negative infinite term $-k_1\omega_1(x_1)x_1^2$ and a bounded term $e_{D1}x_1$. Therefore, the energy function becomes smaller when $x_1$ is at the boundary value, and then the position $x_1$ becomes smaller without crossing the boundary.

Approximately, the position $x_1$ will never exceed the boundary $-320$. The position $x_2$ will also be limited to the range $[-240, 240]$. This completes the bounded stability analysis of the system state and the proof of the constrained system output here.

## 4. Results of Simulation and Experiment

The effectiveness and superiority of the proposed controller based on the GPIO and gain function are to be verified in this section from a simulation and experiment. In the simulation part, the control performance of the gain function-based method is demonstrated compared with the low-gain and high-gain approaches. In addition, for the disturbed system, the proposed method is compared with the one without GPIO to verify the anti-disturbance ability of the system under the situation of small output error. In the experimental part, in order to verify the performance improvement of the system combined with disturbance compensation, low-speed and high-speed rotating targets are tracked by the visual tracking system to simulate the actual situation, where the method without disturbance compensation is compared.

### 4.1. Numerical Simulations

In this section, the control problem of the visual tracking system is investigated to demonstrate the effectiveness of the proposed method. Firstly, in order to verify the advantage of the gain function approach, three different controllers are given for the system without disturbances, which are the proposed method, the high-gain method, and the low-gain method, respectively. Then, for the system with disturbance, the comparison between the proposed method based on GPIO and the same one ignoring GPIO is developed, to verify the improvement of the combination of gain function and GPIO for the anti-disturbance performance of the system.

The nominal values of parameters related to the camera in the system are set as $k = 28$ and $\lambda = 100$. The parameters of the disturbance observer and control law are selected as follows: $l_{11} = 8$, $l_{12} = 6$, $l_{21} = 16$, $l_{22} = 9$, $l_{31} = 64$, $l_{32} = 27$, $k_1 = 0.3$, $k_2 = 0.2$. The feedback gains of the high-gain and low-gain methods are selected as $k_{h1} = 2$, $k_{h1} = 1.5$, $k_{l1} = 0.3$, $k_{l1} = 0.2$, respectively. The criterion of parameter selection is to explain the

instability and noise amplification of the high-gain method, with the slow response and low disturbance suppression ability of the low-gain method.

The following results can be found from Figure 3. The high-gain method causes system fluctuation. Even in the case of a small output error, it produces a large control effect to cause the system to return to the equilibrium point. Although the low-gain approach has better noise suppression ability in practical engineering, its slow response makes it difficult for the system to track the target in time, especially when the target moves quickly. Obviously, the controller based on the gain function has the best control performance, which cannot only achieve fast convergence with large error, but also has good noise suppression with small error. According to Figure 4, the control inputs generated by the three methods also reflect the internal law of the controllers.

Because the disturbance in this system cannot be ignored, the constraint controller based on GPIO is of great significance. As shown in Figure 5, since the GPIO method actively estimates the disturbance and compensates for it in the controller, this is a feed-forward idea. The proposed controller has a good anti-disturbance effect, compared with the controller ignoring GPIO. We find that the control inputs generated by the two methods are similar according to Figure 6, which also ensures the fairness of the comparison.

### 4.2. Experimental Results

In the experimental part, the effectiveness and superiority of the proposed control method are verified on a two-axis ISP. The experimental test includes the following three cases. In case 1, when the disturbance is very small due to the stationary target, the traditional high-gain and low-gain methods are compared to show the advantage of the gain function method. In case 2, a medium-speed moving target is regarded as a tracking task. The proposed control law combining the GPIO and gain function is verified to have satisfactory control performance, compared with the same method without GPIO. In case 3, the high-speed moving target is tracked to show the output constraint ability of the proposed control approach, which is compared with the fixed-gain controller with GPIO.

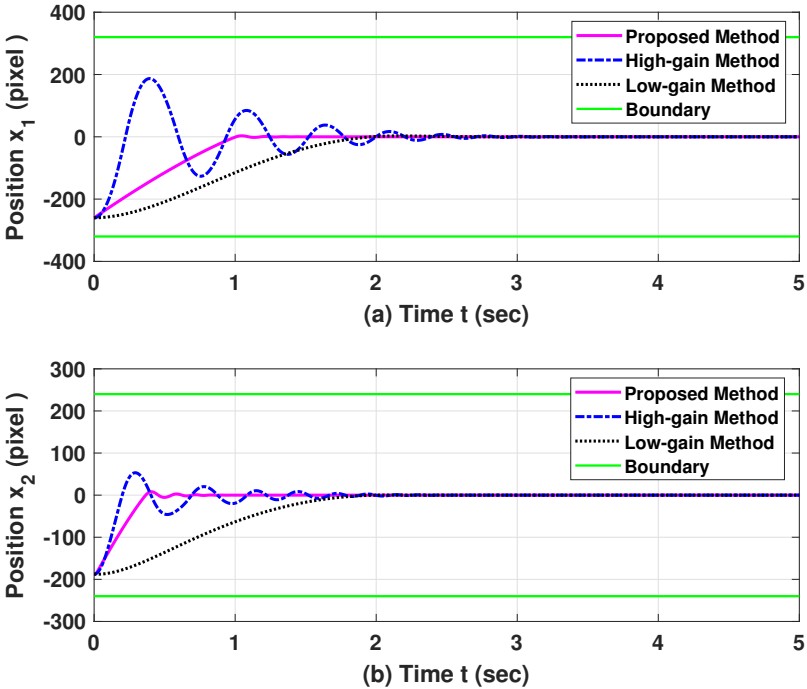

**Figure 3.** Step response curves of undisturbed system under three methods in simulation part: (**a**) azimuth-axis target position, (**b**) pitch-axis target position.

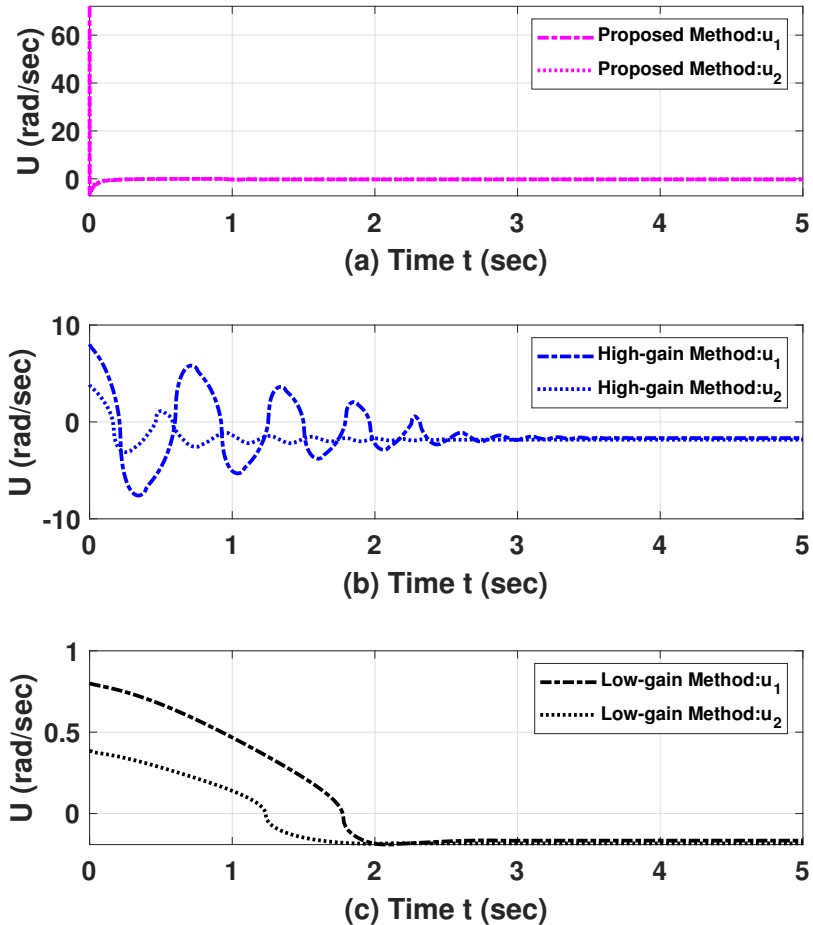

**Figure 4.** Control inputs of undisturbed system under three methods in simulation part: (**a**) control input under proposed method, (**b**) control input under high-gain method, (**c**) control input under low-gain method.

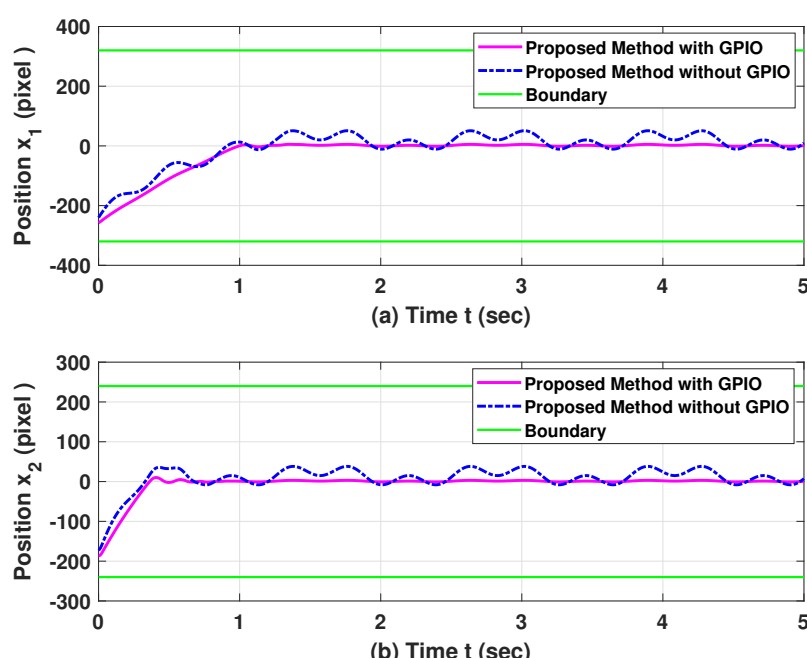

**Figure 5.** Step response curves of disturbed system under two methods in simulation part: (**a**) azimuth-axis target position, (**b**) pitch-axis target position.

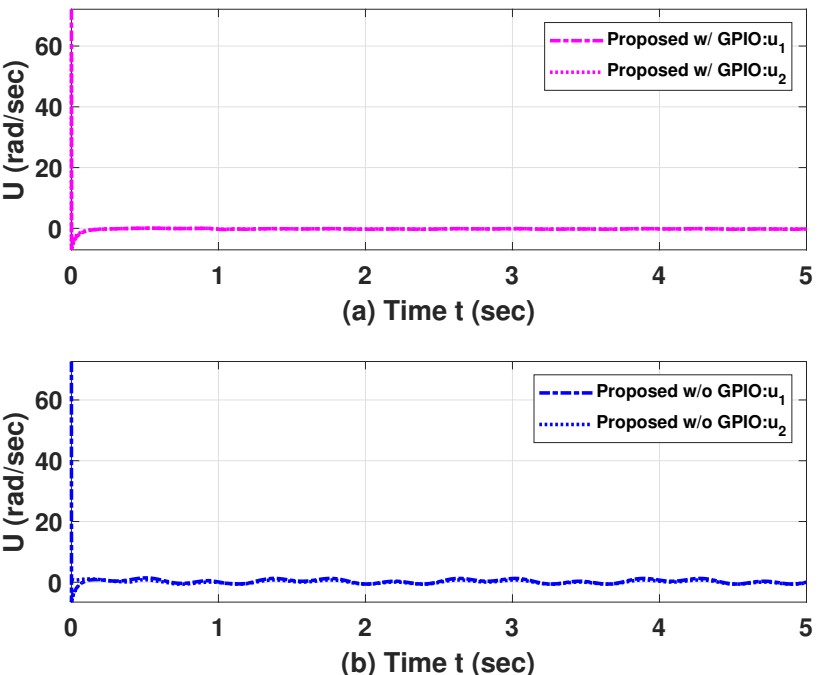

**Figure 6.** Control inputsof disturbed system under two methods in simulation part: (**a**) control input under proposed method with GPIO, (**b**) control input under proposed method without GPIO.

### 4.2.1. Introduction to Experimental Equipment

The equipment of the system in this section includes a two-axis ISP and a moving target, as shown in Figure 7. The two-axis ISP system comprises pitch and yaw gimbals, a color camera, and an inertial angular velocity measurement unit: a fiber optic gyroscope (FOG), simulated carrier, and the related software. The pitch and yaw angular velocity of the camera in inertial space are measured by the FOG (VG091A) with an acquisition frequency of 40 kHz. The position information of the target in the image is obtained via a color camera (DFK21BU04) with an acquisition frequency of 20 Hz. Moreover, the moving target (a black cross sign) is driven by servomotor equipment, the rotation rate of which is controllable to simulate a variety of actual target motion forms.

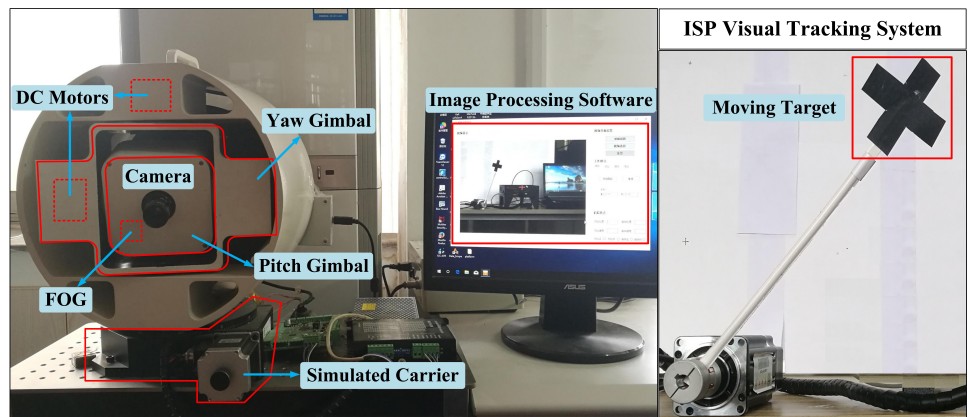

**Figure 7.** Physical depiction of the experimental equipment of the ISP visual tracking system.

### 4.2.2. Case 1: Step Response to Target with Large Initial Output Error

For the high-gain and low-gain control methods, due to their fixed feedback gain, a satisfactory control effect is difficult to guarantee in the whole process of system operation. The approach based on the gain function solves this problem well; that is, it produces a greater control effect in the case of large error to quickly recover the system, and adopts

a smaller feedback gain in the case of small error to avoid noise amplification. In the experiment, the parameters of the controller and disturbance observer are the same as those in the simulation part.

The following conclusions can be found from Figure 8. The high-gain method has a faster response time; however, it always maintains a too high feedback gain, which brings great fluctuation to the system near the equilibrium point. The convergence time of the low-gain control approach is too long, and the anti-disturbance ability with small error cannot be guaranteed, which are fatal disadvantages in a visual tracking system. Obviously, the controller based on the gain function achieves the best control performance.

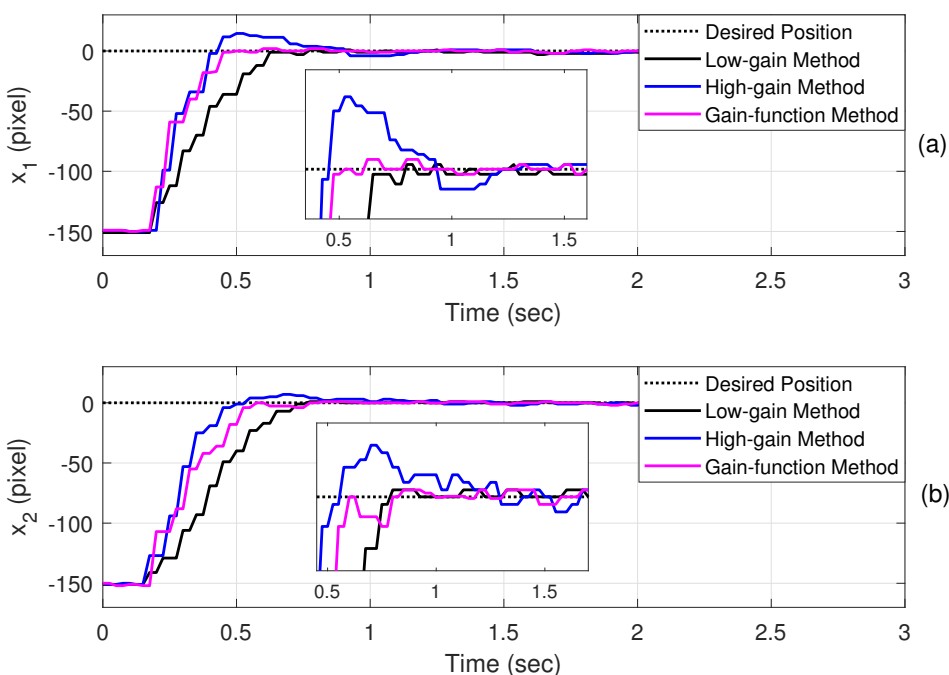

**Figure 8.** Step response curves of three methods under large initial output error in Case 1: (**a**) azimuth-axis target position, (**b**) pitch-axis target position.

### 4.2.3. Case 2: Anti-Disturbance Ability of the System

An important contribution of this paper is to introduce GPIO to improve the anti-disturbance ability of the system with small error. In a situation with a large output error, the huge feedback gain of the proposed method will bring a sufficient control effect to suppress the disturbance. Therefore, the medium-speed target is selected as the task of this case to highlight the role of GPIO in the case of small error. A gain function-based controller without GPIO is used as a comparison. In order to ensure the fairness of the experimental comparison, the same controller parameters are selected for both approaches.

According to Figure 9, it can be found that the anti-disturbance effect of GPIO is very obvious and necessary. The visual tracking system of an ISP is subject to multi-source disturbances in actual operation, which greatly reduce the tracking accuracy of the system. Thanks to the active disturbance estimation and compensation, the proposed control method shows strong robustness against the time-varying disturbance.

### 4.2.4. Case 3: Output Constraint Ability of the System

The output constraint ability of the proposed method is verified in this case. In order to make the system work near the boundary, a fast-moving target is tracked in this task. The high-gain feedback control has a certain output constraint ability, so it is selected as the comparison method. The high-speed target is also a large disturbance of the system; therefore, the GPIO method is combined with the gain function-based controller. In order

to ensure the fairness of the comparison, the GPIO method is also combined with the high-gain controller.

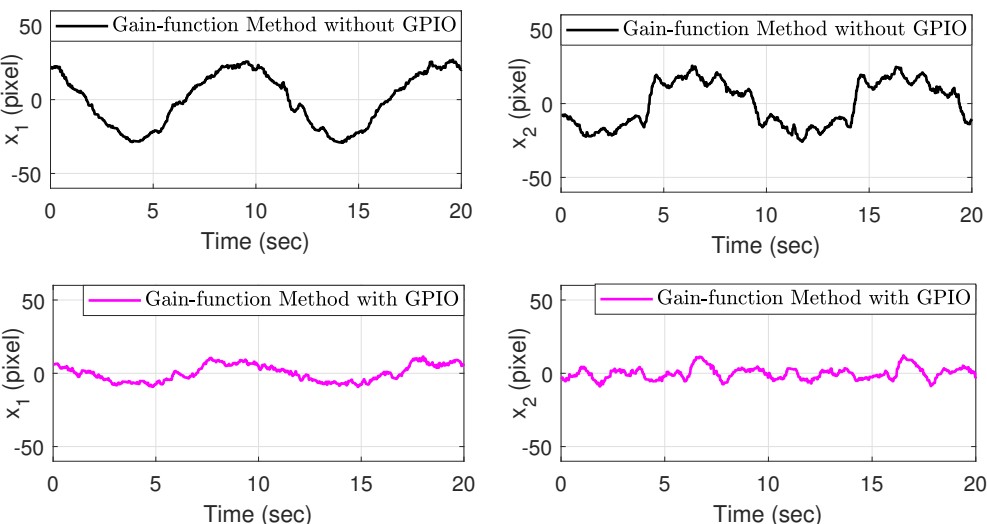

**Figure 9.** Response curves of tracking moving target under two methods in Case 2.

Because the gain function can produce a feedback gain approaching infinity near the boundary, the system has a sufficient control effect to adjust the position of the target, so as to ensure that the target is always maintained in the field of view of the camera. Since the feedback gain of the high-gain method is fixed, in order to balance the steady-state output of the system with small error, the gain cannot be set too large, resulting in a limited control effect near the boundary. As shown in Figure 10, although neither method leads to target loss for the sake of experimental integrity, the output errors are significantly different. The proposed method based on the gain function causes the target to be situated far away from the boundary. Therefore, the proposed method can strictly realize the system output constraints.

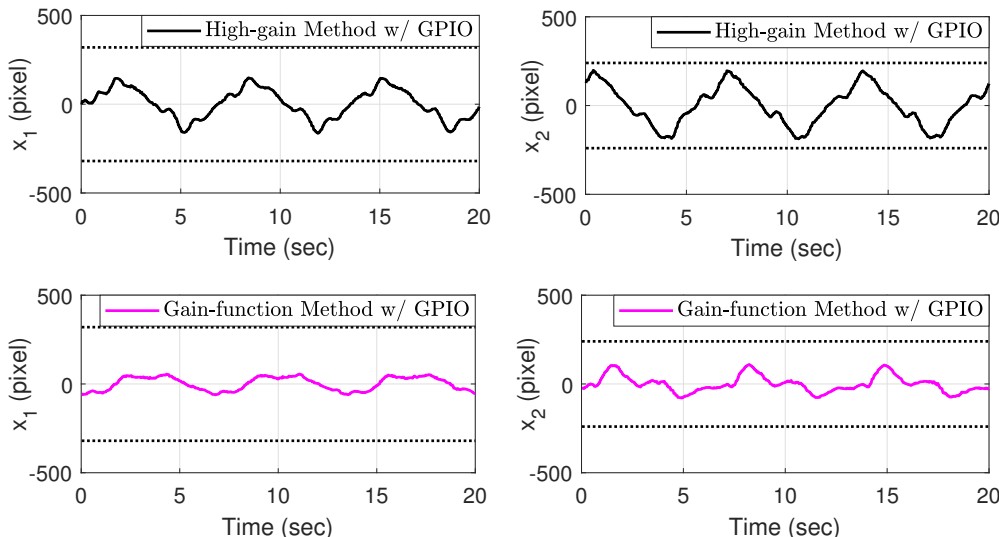

**Figure 10.** Response curves of tracking high-speed target under two methods in Case 3.

## 5. Conclusions

In order to solve the target loss problem of the visual tracking system of an ISP affected by disturbances, a control algorithm based on a gain function and GPIO has been

proposed in this paper. Firstly, the model of the system considering disturbances and output constraints has been established. Then, to avoid the tracked target being lost in the field of view of the camera, the control method based on the gain function technique has been designed to ensure that the output is constrained in the image plane. Moreover, the active disturbance rejection method has been introduced to improve the anti-disturbance ability of the system with a small output error, obtaining satisfactory tracking performance. The stability analysis and the proof of the constrained output have been given following the controller design. Finally, the results of simulations and experiments have been shown to illustrate the promised advantages of the proposed control approach. Furthermore, the finite-time output constraint control approach of a system with disturbance is to be studied to achieve faster convergence performance.

**Author Contributions:** Conceptualization, X.L.; methodology, X.L., P.Q.; formal analysis, X.L.; data curation, X.L., P.Q.; writing—original draft preparation, X.L.; writing—review and editing, J.Y., P.Q.; supervision, J.Y.; project administration, J.Y.; funding acquisition, J.Y. All authors have read and agreed to the published version of the manuscript.

**Funding:** This research was funded in part by the National Natural Science Foundation of China under Grant 61973080 and in part by the Shenzhen Science and Technology Innovation Committee (STIC) under Grant JCYJ20190813152603594.

**Conflicts of Interest:** The authors declare no conflict of interest.

## Abbreviations

The following abbreviations are used in this manuscript:

| | |
|---|---|
| ISP | Inertial stabilized platform |
| BLF | Barrier Lyapunov function |
| GPIO | Generalized proportional integral observer |

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
