# Peer review of "Gain Function-Based Visual Tracking Control for Inertial Stabilized Platform with Output Constraints and Disturbances"

_electronics, doi:10.3390/electronics11071137_

Round 1

Reviewer 1 Report

The authors combine the disturbance observer and the variable gain function method to solve the target loss problem for the ISP visual tracking system with the output constraints and disturbances. 

The methods used are appropriate, and the results of the experiments support the promised advantages of the approach adopted.  

Thus, I have no amendments to suggest.

Author Response

Thanks a lot for your suggestions and comments on our works. We have made necessary changes marked in blue in the revised version.

Reviewer 2 Report

A composite control strategy was designed to deal with output constrains1
and disturbances of the visual tracking system for an inertial stabilized platform, which combines the active disturbance compensation and the variable gain function technique.It seems to me that the topic of the manuscript shows its own interest.However, the authors should pay attention in correcting the typsos, extend the introduction and present a much better motivation for typing their work.In addition, new a nd related references should be added.

Author Response

We would like to express our sincere thanks for your insightful comments and constructive suggestions. The manuscript has been revised by fully following your comments. Please note that the changes have been highlighted in blue. Here are three detailed responses to your comments.

Comment 1:The authors should pay attention in correcting the typos.

Response 1:Thanks a lot for your insightful suggestion. We have thoroughly checked the writing of the manuscript and corrected the inappropriate expressions. 

Comment 2:Extend the introduction and present a much better motivation for typing their work.

Response 2:We thank the reviewer very much for his/her useful comment. We are sorry for not stating the motivation clearly in the previous submission. The introduction has been reorganized according to your suggestions. In the visual tracking system of an inertial stabilized platform, it is a key problem to prevent the target from losing in the field of view of the camera. The method based on gain function is applied to realize the system output constraint, because it exerts enough control effect when the output is close to the boundary. However, when the system works near the equilibrium point, the gain function produces a small feedback gain, resulting in unsatisfactory anti-disturbance performance. Therefore, the proposed control method based on disturbance observer and gain function is to deal with the output constraints and disturbances in the system. This motivates the research work of this paper.

Comment 3:In addition, new and related references should be added.

Response 3:Thank you very much for the constructive suggestion. In order to enrich the content and improve the quality of the introduction, six relevant references in the past three years have been added, which have been  highlighted in blue in the revised version.

Reviewer 3 Report

Report

on the paper “Gain Function Based Visual Tracking Control for Inertial Stabilized Platform with Output Constraints and Disturbances” by Xiangyang Liu, Jun Yang, and Pengyu Qiao.

This paper is devoted to the study of very interesting and important applied problem – target loss problem of the visual tracking system of an inertial stabilized platform (ISP) affected by perturbations (disturbances).

To realize the high-performance control for the ISP visual tracking system with the output constraints and disturbances, the authors proposed a composite control strategy, which combines the disturbance observer and the variable gain function method. This method includes modelling of the visual tracking system of an ISP that considers multi-source disturbances where the controlled output is the constrained position of the target in the image plane. Then, it was designed the GPI observer for compensation of the lumped disturbance in the system, and the variable gain function was introduced in controller to ensure that the system output is always in the given range. The basic question – stability of the proposed method, is given theoretically that has a special importance.

The simulation and experimental results illustrate the advantages of the gain function based approach and the proposed composite control method. Obtained results (Theorem 1, Theorem 2) are of big theoretical interest for the design of the composite control strategy and the stability analysis.

By my opinion, the paper “Gain Function Based Visual Tracking Control for Inertial Stabilized Platform with Output Constraints and Disturbances”  

                 - is original and doesn’t contradict to ethical or policy issues;

- the question posed by authors is new and well defined;

- the methods used by authors are appropriate;

- the data are sound and well controlled;

- the discussion and conclusions are well balanced;

- the title and abstract convey the obtained results;

- the writing is acceptable;

contains good scientific results and can be published in Electronics.

I declare that I have no competing interests.

Referee.

Author Response

Many thanks for your valuable time on reading and reviewing our manuscript. We have made necessary changes marked in blue in the revised version.